# Ultrafast Resolution-Enhanced Digital Optical Frequency Comb-Based BOTDA with Pump Pulse Array Coding

**DOI:** 10.3390/s20226411

**Published:** 2020-11-10

**Authors:** Yichang Wu, Chengkun Yang, Jingshun Pan, Qi Sui, Dawei Wang

**Affiliations:** 1School of Electronics and Information Technology, Sun Yat-sen University, Guangzhou 510275, China; wuych27@mail2.sysu.edu.cn (Y.W.); panjsh3@mail2.sysu.edu.cn (J.P.); wangdw9@mail.sysu.edu.cn (D.W.); 2Southern Marine Science and Engineering Guangdong Laboratory (Zhuhai), Zhuhai 519000, China; suiqi@sml-zhuhai.cn

**Keywords:** Brillouin, ultrafast, distributed sensing, pump pulse array, resolution enhancement, SNR

## Abstract

In this letter, a resolution enhancement and signal-to-noise ratio (SNR) improvement scheme for digital optical frequency comb (DOFC)-based Brillouin optical time-domain analysis (BOTDA) ultrafast distributed sensing employing a pump pulse array is proposed. Based on the properties of the time-invariant linear system and the cyclic revolution theorem, experimental results indicate that its spatial resolution reaches 10.24 m while the frequency uncertainty is below 2 MHz over a 9.5 km fiber. Moreover, the response time is only 209.6 μs and the temperature measurement error is less than 0.52 °C.

## 1. Introduction

In recent years, Brillouin optical time-domain analysis (BOTDA)-based distributed optical fiber sensing has become the subject of numerous studies for its highly precise temperature and strain-distributed measurement accuracy [1,2,3,4]. For conventional BOTDA systems, the frequency difference of the pump-probe scanning process is used to reconstruct the Brillouin gain spectrum (BGS) and Brillouin frequency shift (BFS), which can be linearly mapped to temperature and strain distributions along the fiber. It normally takes a few minutes to complete the single-time measurement of the entire fiber link due to the time-consuming frequency scanning process and the trace averaging algorithm, dropping off the distributed dynamic information of the sensing fiber [5,6].

To eliminate these two major factors that consume the sensing time, several sweeping-free techniques have been proposed to realize fast and dynamic measurements, such as slope-assisted BOTDA [7,8] and ultrafast sweeping BOTDA [9,10]. Besides, digital optical frequency comb (DOFC)-based BOTDA [11,12,13] and chirp-chain-based BOTDA [14,15,16] are presented, which are able to reconstruct the BGS and locate the BFS with one single shot. The combination of DOFC-BOTDA and multiple pump pluses provides a better signal-to-noise ratio (SNR) and breaks the inherent limitation between the spatial resolution and the detection accuracy within 0.1 ms [17]. The chirp-chain-based BOTDA utilizing the Brillouin loss scheme achieves 150 km long range fiber sensing within a few seconds [14].

However, the single-shot or fast-scanning BOTDA is confronted with several questions. First, renouncement of frequency scanning and time average severely increases the random noise on the measured BGS, decreasing the SNR of the signal and precision of the frequency [12]. Furthermore, short measurement time means much less effective information, suffering poor spatial resolution, sensitivity, or short sensing range [12,17]. Besides, the multiple-pump-pulses scheme requires an identical power of pulses, which is difficult to achieve with a high peak power of pulses [17]. Some of the fast sensing schemes can provide a fine resolution or long sensing distance by a large amount of averaging, leading to a long measuring time (a few seconds) and missing the dynamic information [14].

To overcome the challenge mentioned above, this paper starts with an introduction of a new resolution and SNR enhancement scheme without lengthening the measurement time in Section 2, combining the DOFC-BOTDA with pump pulse array coding, which can cut down the frequency uncertainty while maintaining the spatial resolution. In addition, the simulation test of the proposed sensing scheme is demonstrated in Section 3, by comparing the sensing results between our proposed scheme and the conventional counterpart. In Section 4, the experimental setup and the data processing method are introduced, and the experimental performances of the proposed sensing scheme are evaluated. Finally, some conclusions and a summary are given in Section 5.

## 2. Sensing Principle

In the proposed scheme based on scanning-free BOTDA, a continuous wide-band DOFC signal is used as the probe to detect and acquire the BGS (Brillouin gain spectrum) with a single shot. The DOFC probe with single- and multiple-pump BOTDA has been proposed for high-speed sensing and dynamic measurement [11]. Figure 1 shows the sensing principle of the DOFC-based BOTDA.

In this paper, the pulse array is used as the pump to increase the signal-to-noise ratio and enhance the detection accuracy. The pulse array is a periodic sequence of a NRZ PRBS (Non-return-to-zero pseudo-random binary sequence) modulated by a sinusoidal wave. The optical probe and pump signals are transmitted in opposite directions. The BGS obtained by probe waves is the superposition of the BGS generated by the simulated Brillouin scattering of each pulse.

Intensity variations in the probe wave induced by stimulated Brillouin scattering can be expressed as [18]
(1)$$\Delta I_{\mathrm{CW}}(t,\Delta \nu )=I_{CWL}e^{-\alpha L}\left\{\exp\left(\int_{v_{g}t/2}^{v_{g}t/2+\Delta z}g_{B}(\xi ,\Delta \nu )I_{P}(\xi ,\Delta \nu )d\xi \right)-1\right\}$$
where $\Delta I_{\mathrm{CW}}$ is the intensity fluctuation measured at the near end of the fiber z=0 as a function of time t and frequency offset between pump and probe light Δν. $I_{CWL}$ is the input probe intensity at z=L, L is the length of the fiber, α is the loss of optical fibers, $v_{g}$ is the group velocity, $g_{B}(\xi ,\Delta \nu )$ and $I_{P}(\xi ,\Delta \nu )$ are the Brillouin gain coefficient and pump intensity at z=ξ, respectively. The short simulated Brillouin scattering (SBS) interaction length taking place in BOTDA sensors leads to a very small Brillouin gain, allowing us to linearize Equation (1) as
(2)$$\Delta I_{\mathrm{CW}}(t,\Delta \nu )\propto \int_{v_{g}t/2}^{v_{g}t/2+\Delta z}g_{B}(\xi ,\Delta \nu )I_{P}(\xi ,\Delta \nu )d\xi$$

If $I_{P}(\xi ,\Delta \nu )$ is no longer a single pulse, the entire interaction length should be taken into account in the upper limit of the above integration. From Equation (2), the SBS process can be regarded as a linear time-invariant, as well as frequency-invariant, system. As for conventional linear time-invariant (LTI) systems, the linear convolution theorem tells us that the product of the Fourier transform of the impulse response function and input function in the time domain is equal to the Fourier transform of the output response function. In our case of the SBS process, it can be extended to two dimensions (2D), i.e., the frequency–time domain. The 2D Fourier transform of the output response function should be the product of the Fourier transforms of the input function and impulse response function, i.e.,
(3)$${ℱ}_{2D}\left\{I\left(t,\nu \right)\right\}\cdot {ℱ}_{2D}\left\{R\left(t,\nu \right)\right\}={ℱ}_{2D}\left\{O\left(t,\nu \right)\right\}$$

Input function I, impulse response function R, and output response function O are functions of time t and frequency ν. In real applications, as sampling is discrete, a 2D function in the frequency–time domain is converted to a discrete 2D matrix. In addition, the linear convolution theorem should be replaced by the cyclic convolution theorem [19], i.e., the product of the discrete Fourier transform of the impulse response matrix and input matrix in the time–frequency domain is equal to the discrete Fourier transform of the output response matrix.
(4)$$I_{F}\left(x_{n}{,y}_{m}\right)={ℱ}_{2D}\left\{I\left(t_{n},\nu_{m}\right)\right\},R_{F}\left(x_{n}{,y}_{m}\right)={ℱ}_{2D}\left\{R\left(t_{n},\nu_{m}\right)\right\},\;O_{F}\left(x_{n}{,y}_{m}\right)={ℱ}_{2D}\left\{O\left(t_{n},\nu_{m}\right)\right\}$$
$$I_{F}\left(x_{i}{,y}_{j}\right)\cdot R_{F}\left(x_{i}{,y}_{j}\right)=O_{F}\left(x_{i}{,y}_{j}\right),i=1,2\ldots N,j=1,2\ldots M$$
where the size of all matrices is N∗M. In this case, the product of the matrix is the Hadamard product, as presented in Equation (4). The output response of the system is detectable, while the impulse response matrix implies that the output of the system with a single pump pulse should be like the conventional BOTDA as we expect. By designing the input response matrix *I*, detecting the output response matrix *O*, and performing the inverse operation of Equation (4), the impulse response matrix *R* is solvable, as depicted in Figure 2. Compared to the BOTDA systems with single or several pump pulses, the quasi-continuous pump pulse array is able to improve the SNR of the system.

In Figure 2, two dimensions of matrices represent time and frequency. The input matrix of the system, pump matrix, is carefully designed. In this paper, we choose PRBS as the code of the pump array, while bits 1 and 0 indicate that a single pump pulse is on or off, respectively. Every pump pulse here is modulated by a sinusoidal wave with an identical frequency. Therefore, the elements of the pump matrix are 0 except those in the corresponding row of frequency, as shown in the first row of the pump matrix in Figure 2. The output matrix is constructed by splitting the received signal into independent frames, performing FFT on each frame and aligning the FFT results. After the construction of the input and output matrix, 2D-FFT is performed to change the matrix into the “frequency” domain (the frequency generated by 2D-FFT, not the real frequency). The product on the left side of Equation (4) is the Hadamard product instead of matrix multiplication. In order to recover the impulse response matrix, the inverse operation of Hadamard multiplication should be applied. Divided by the input matrix element by element, the output matrix becomes the impulse response matrix in the frequency domain. After a two-dimensional inverse fast Fourier transform (2D-IFFT) on the result, the impulse response matrix is recovered.

## 3. Simulation

We have simulated the Brillouin sensing process for a 1.6 km single-mode fiber (SMF). The length of one DOFC frame is 200 ns, corresponding to 20 m spatial length, while the frequency step is 5 MHz according to the inherent restriction imposed by the length of a DOFC frame and the frequency spacing after the FFT, i.e., the spatial and frequency resolution. Figure 3a shows the simulated impulse response matrix without noise by a 12.09 GHz sinusoidally modulated pump pulse input. Each column in this matrix represents a DOFC frame, carrying a Lorentzian-shaped BGS. The full-width at half-maximum (FWHM) of the BGS is set to 30 MHz. The typical BFS of this fiber is 10.8 GHz, resulting in the BGS being centered at 1.29 GHz on the DOFC probe. In addition, there are two irregular parts with different BFSs. One is located at 980–1000 m with a 10.96 GHz BFS, while another is at 1280–1320 m with a 10.84 GHz BFS. According to the setting frequency of the pump pulse, their central frequencies of the BGS should be located at 1.13 and 1.25 GHz, respectively.

Here, we choose PRBS9 (511 bits) as the pump array in the time domain using 11.09 GHz sinusoidal wave modulation, while bits 1 and 0 indicate that a single pump pulse is on or off, respectively. The length of each bit in the pump array is equal to a DOFC frame (200 ns in this case). The PRBS can provide a perfectly flat intensity plane without extreme points in the frequency domain after a discrete 2D Fourier transform, which can avoid the severe accuracy issue during the calculation and become robust. Without a flat intensity plane in the frequency domain, the quality of the solved impulse response matrix will deteriorate rapidly as the power of noise in the output response matrix increases. As there is only one single frequency in this pump array, all the elements of the pump matrix should be 0 except the first line that represents the 11.09 GHz sinusoidally modulated PRBS9. Although the pump array is periodic, the pump matrix only records one period as the cyclic convolution regards it as a periodic signal already. The size of the pump matrix should be expanded to 511 × 511 to ensure that the intensity remains flat after the 2D fast Fourier transform (2D-FFT).

Figure 3b shows the simulated output signal matrix of the sensing system in the presence of the pump matrix input. It is generated by overlapping the impulse response matrix that each single pulse in the pump matrix provides. As the pump array is periodic, the output signal should have the same cycle period in the time domain. Besides, white Gaussian noise is added on the output signal matrix, and the SNR is set to 20. Figure 3d,e respectively show the calculation results with the pump array input and a single pump pulse input under the same SNR. The BGS is more obvious using the pump array input than the single pump pulse input. It is an efficient way to improve the SNR of the received BGS and reduce the BFS uncertainty. Figure 3f demonstrates the comparison of the trace of the BGS peak along the fiber. Background noise is significantly suppressed using the pump pulse array input, with an SNR enhancement of about 5 dB.

## 4. Experiment, Data Processing, and Discussion

The experimental setup of the proposed BOTDA sensing system is depicted in Figure 4. A tunable continuous-wave laser (Keysight N7714A) provides a 1550 nm beam with a 100 kHz linewidth. A 90/10 optical coupler is employed to split the 1550 nm laser into two branches for the pump and probe signal.

As for the pump side, an arbitrary waveform generator (AWG, Keysight M8195A, Santa Rosa, CA, US) gives an electric pulse array arranged as PRBS11 modulated with 800 MHz sinusoidal waves. Each bit in this array lasts for 102.4 ns. For the first half of a bit (about 50 ns), the signal turns to the zero level in the case of 0, whereas it turns to a sinusoidal wave in the case of 1. However, it always turns to the zero level at the latter half of the bit in order to avoid the overlapping issue of orthogonally polarized pulses mentioned later. Then, the pulse array is upconverted to 11.1 GHz by a mixer and a sinusoidal wave generated by a signal generator, and goes through a microwave bandpass filter to filter out the superfluous frequencies induced by the mixing process. The filtered pulse array is imposed on the pump light via a high-extinction-ratio Mach–Zehnder modulator (MZM, iXblue MXER-LN-20, Saint-Germain-en-Laye, France) biased at the null point. The achieved extinction ratio of carrier suppression exceeds 30 dB in order to avoid the four-wave mixing phenomenon in optical fibers. After being amplified by an erbium-doped optical fiber amplifier (EDFA), the lower sideband and remaining carrier are filtered out by a fiber Bragg grating (FBG) with 10 GHz 3 dB-bandwidth. The second EDFA amplifies the remaining upper sideband signal. To eliminate the influence of the polarization fluctuations to the SBS effect, a scheme of orthogonally polarized pump pulses is used here [20]. A polarizing beam splitter (PBS) is used to split the optical pump array into two branches with an orthogonal polarization state. Two polarization controllers (PC) are set before the PBS and one of the branches to adjust the polarization state, ensuring both branches get identical optical power. Besides, one of the branches is delayed by an SMF, in which the pump array can go for a 51 ns delay, about half the length of one bit of the PRBS array. By using a polarizing beam combiner (PBC), two branches are combined to generate an orthogonally polarized optical pump pulse array. As the effective signal only occupies the front half of the bits in the original pump array, delay and recombination will avoid the overlapping of the two orthogonally polarized optical signals. Due to the orthogonally polarized pump pulses with identical power, regardless of the direction of polarization of the probe light, the Brillouin gain is stable. Furthermore, a programmable wave-shaper is used to reshape the optical pump array and obtain a higher extinction ratio. Finally, the array is amplified by EDFA3 and launched into a fiber-under-test (FUT) of 9.5 km in length via an optical circulator. The measurement for one single test of the entire fiber link is equal to one period of the pump array, i.e., 209.6 μs.

As for the probe side, another channel of the AWG gives a periodic PRBS as a digital electrical frequency comb (DEFC) signal. Each frame of the DEFC lasts for 102.4 ns, which is as long as the length of one bit in the pump array, corresponding to a frequency spacing of 9.77 MHz and spatial resolution of 10.24 m. The light is modulated by this DEFC signal via another MZM (Fujitsu FTM7937EZ) biased at the linear operating point to generate the baseband DOFC probe signal. After being amplified by EDFA4, the probe signal passes through an optical isolator and an optical circulator, which would filter out the pump signal from the opposite side, and finally, the probe signal is sent into the FUT.

At the receiver end, the probe signal is detected by a photodiode (PD). The output signal is transmitted into a real-time oscilloscope (OSC) with a 20 GS/s sampling rate. After data acquisition, digital signal processing (DSP) is performed to solve the impulse response matrix. The DSP is shown as the flowchart in Figure 5a.

The DOFC signal without the BGS in the frequency domain is shown in Figure 5b as a background reference noise. By synchronization of the received signal, the start points of all DOFC frames can be located. By dividing the continuous received signal into independent frames, fast Fourier transform (FFT) and background denoising could be used to obtain the Brillouin gain information carried by each frame. As the pump array is periodic, the experimental received output signal should also be periodic in the time dimension, as shown as Figure 5c. Then, smoothing in the frequency dimension would help to reduce the glitch on the signal. Furthermore, each frame is up-sampled by a factor of four in the frequency domain to achieve a narrower frequency spacing of 2.44 MHz. The generation of the pump matrix has been mentioned above, and the output matrix is generated by using a suitable data window onto the up-sampling signal. Here, the size of the pump matrix and the data window is 2047 × 201. In addition, by performing 2D-FFT on both matrices, dividing element by element, and conducting two-dimensional inverse fast-Fourier-transform (2D-IFFT) retrieval to the time–frequency (or space–frequency) domain, the impulse response matrix can be solved. In order to align the start point of the FUT to the zero point in the space dimension of the matrix, it is necessary to introduce a proper shift in the data window on the up-sampling signal. The impulse response matrix is shown in Figure 5d, where the time dimension has been changed into the space dimension (10 ns corresponds to 1 m based on the refractive index of optical fiber of 1.5) and the probe frequency has been changed into the BFS by being subtracted from the pump frequency of 11.1 GHz.

To acquire the accurate BFS, we have performed a cross-correlation algorithm and Lorentz fitting. The cross-correlation algorithm is based on the fact that Gaussian white noise is uncorrelated with the Lorentz lineshape while the effective BGS signal is strongly correlated. As the response signal is a linear combination of the BGS and noise, cross-correlation with a standard Lorentz lineshape can be used to denoise the signal [15]. As the correlation of two Lorentz shapes is still a Lorentz shape, iterative correlations can further denoise the signal. However, the correlation will broaden the width of the Lorentz shape and increase the frequency uncertainty, so we take the proper iterative time as two. Besides, by up-sampling, better results can be obtained owing to the fact that a narrower frequency spacing of a standard Lorentz lineshape would allow more effective information. Here, we perform up-sampling by a factor of four. Figure 6a shows a comparison among the signals with no cross-correlation, with a one-time correlation, and with a two-time correlation. The BGS seems asymmetric because of the beating between the upper and lower sideband of the optical probe signal. In addition, after the twice-correlation algorithm, Lorentz fitting can obtain the BFS distribution over the fiber link.

In the temperature test configuration, a segment of 10 m long fibers at 5 km is heated using a thermostat water bath. Figure 6b shows the BGS distribution of the FUT under a room temperature of 25 °C, and Figure 6c shows the BGS distribution when the 10 m fiber section is heated up to 45 °C, where only one frame of the DOFC corresponding to the BFS of the heating part is shifted. Besides, the 10 m fiber section is heated from 25 °C to 60 °C with a 5 °C step, and the measured BFS is shown in Figure 6d. The environment temperature and BFS fit the linear relation, and the maximum deviation from the fitting curve is 0.52 °C. The temperature-BFS ratio is measured to be 0.91 °C/MHz according to the fitting curve. Figure 6e depicts the frequency uncertainty along the FUT through repeated measurement for 10 times under the same environment. The frequency uncertainty is below 1.5 MHz for the first 5 km fiber section. As for the latter 5 km fiber section, though the frequency uncertainty reaches 3.2 MHz when getting close to the end of the fiber link, it is below 2 MHz in most cases, which is much better than those resulting from the inherent restriction between the spatial resolution and frequency uncertainty when using the single pump pulse and DOFC probe, i.e., spatial resolution of 10.24 m with 9.77 MHz frequency uncertainty. As the joints of the optical fibers will provide extra frequency uncertainty, the frequency uncertainty of our proposed this system is below 2 MHz.

## 5. Conclusions

In summary, we have demonstrated an SNR improvement scheme for DOFC-based BOTDA sensing employing a pump pulse array, by which a BFS sensing time of only 209.6 µs could be achieved for a 9.5 km fiber link. By applying two-dimensional cyclic convolution and linear system theory, time as well as frequency, the input power of the system could be greatly improved to obtain better SNR. The spatial measurement resolution reached 10.24 m and the frequency uncertainty was below 2 MHz. Owing to its fast sensing speed and improved spatial resolution, our proposed sensing system is expected to find applications in FPGA-based long-time dynamic measurement.

## Figures and Tables

**Figure 1 sensors-20-06411-f001:**
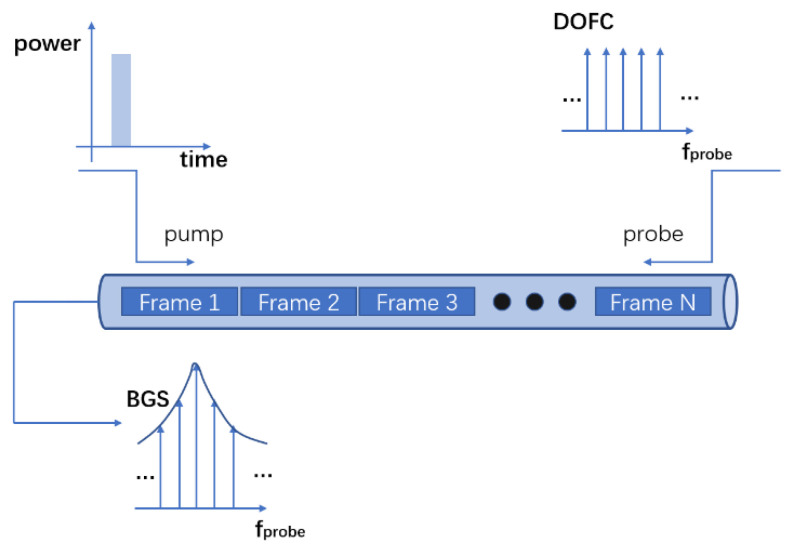
Sensing principle of the digital optical frequency comb-Brillouin optical time-domain analysis (DOFC-BOTDA) with single-pulse input.

**Figure 2 sensors-20-06411-f002:**
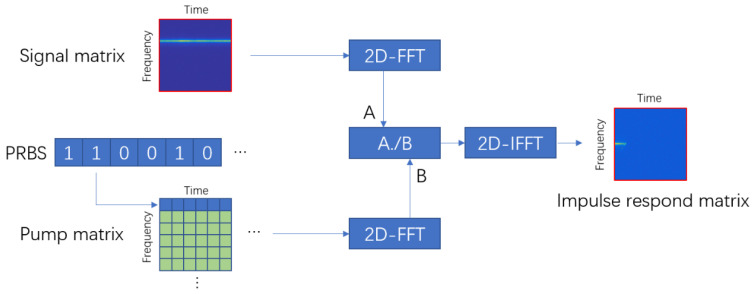
Impulse response matrix calculation procedure.

**Figure 3 sensors-20-06411-f003:**
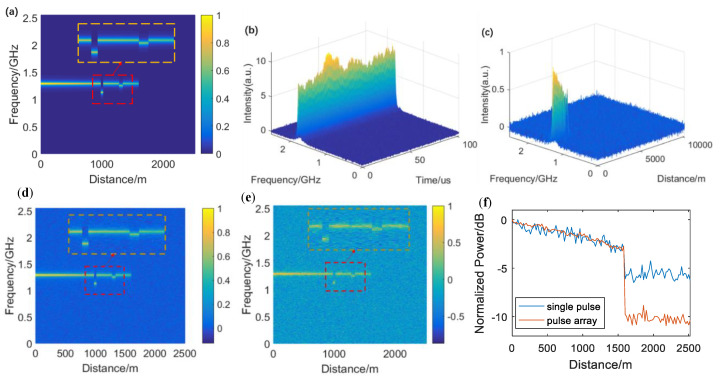
Simulation results. (**a**) Set impulse response matrix without noise, using two irregular Brillouin frequency shift (BFS) points; (**b**) simulated received output signal generated by the sensing system with pump array input; (**c**) calculated impulse response matrix; (**d**) calculated impulse response matrix with pump array input; (**e**) calculated impulse response matrix with single pump pulse input; (**f**) simulated trace for single pump pulse and pump pulse array input case.

**Figure 4 sensors-20-06411-f004:**
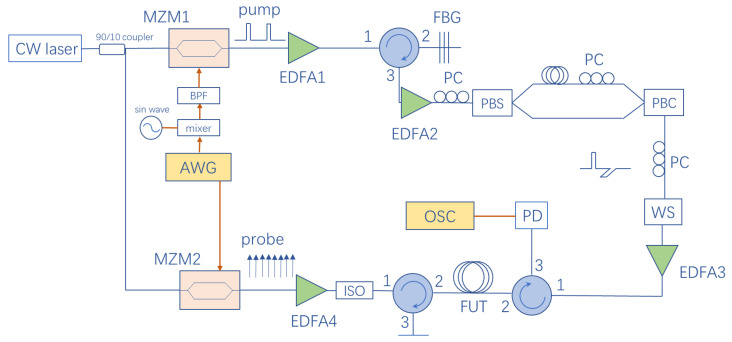
Experimental setup of the proposed BOTDA sensing system. CW laser: Continuous-wave laser; MZM: Mach–Zehnder modulator; EDFA: Erbium-doped optical fiber amplifier; FBG: Fiber Bragg grating; PC: Polarization controller; PBS: Polarizing beam splitter; WS: Wave-shaper; PBC: Polarizing beam combiner; PD: Photodiode; ISO: Optical isolator; FUT: Fiber under test; OSC: Real-time oscilloscope.

**Figure 5 sensors-20-06411-f005:**
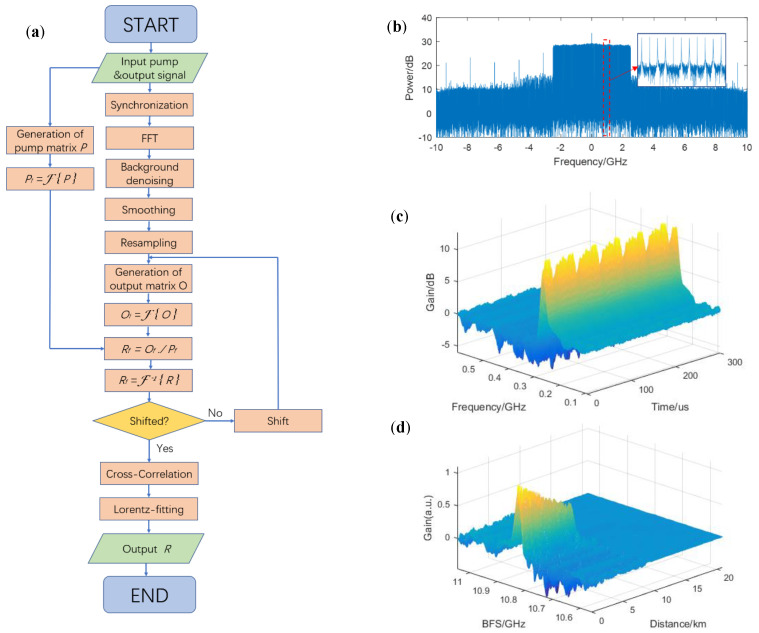
(**a**) Flowchart of digital signal processing; (**b**) DOFC signal in frequency domain; (**c**) experimental periodic received output signal from the sensing system; (**d**) calculated impulse response matrix.

**Figure 6 sensors-20-06411-f006:**
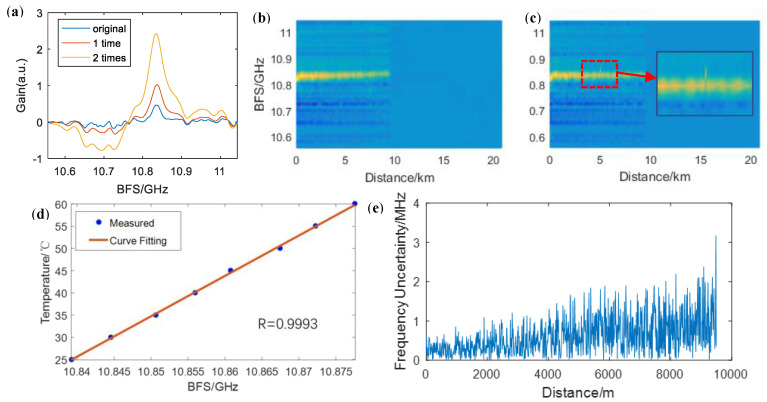
(**a**) Results of cross-correlation with Lorentz lineshape; (**b**) Brillouin gain spectrum (BGS) distribution in normal environment; (**c**) BGS distribution with a 10 m fiber section heating; (**d**) temperature-BFS measurement results; (**e**) frequency uncertainty over the fiber link.

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
