# Peer review of "Ultrafast Resolution-Enhanced Digital Optical Frequency Comb-Based BOTDA with Pump Pulse Array Coding"

_sensors, 2020, doi:10.3390/s20226411_

Round 1

Reviewer 1 Report

The paper looks interesting and succeds in showing improvements in DOFC-based BOTDA, which is much faster than traditional implementation but suffers from intrinsic limitations in spatial resolution.

I found some difficulties in understanding the process of determining the impulse response matrix, which is not well supported by figure 2 (in the figure B and A are input and output, which are called I and O in equation4). Also, the composition of the full pump matrix used in the measurement is not clear: should it be the PRBS repeated on each line, each with a different frequency?

It is also not very clear (despite being correct) how the resulting BGS of the three fiber sections is centered around 1.29, 1.13 and 1.25 GHz respectively. (lines 105-111).

Also, at line 219, the meaning of 1-time and 2-times correlation should be better explained.

In my opinion, the paper can be published provided that the issues I pointed out are well clarified.

Author Response

Thank you for your opinions for our work.

1.The equation (4) shows the relationship among the input, output and impulse response matrix after 2D-FFT (or we can say in frequency domain, although this "frequency" is generated by 2D-FFT instead of the real frequency). For a linear invariant system, the product of input and impulse response matrix is equal to output matrix in frequency domain. Distinguished from matrix multiplication, the product on the left side of equation is performed by multiplying element by element. It is simple to perform an inverse operation, i.e. divide element by element, to obtain the impulse response matrix in frequency domain, which is noted as shown "A./B" in figure 2. More details have been added in part 2 (line 102-114).

2.In simulation, the frequency of pump pulse is set as 12.09 GHz instead of 11.09 GHz. Since the Brillouin scattering process will down-converse the frequency of light, the power of probe light set at low frequency will increase. The Brillouin frequency shift (BFS) in normal environment is set as 10.8 GHz, thus the frequency of center of gain profile on probe light is 12.09-10.8=1.29 GHz. While the BFS of the irregular sections of fiber (like being heated or stressed) are set as 10.96 GHz and 10.84 GHz, their gain profiles are centered at 1.13 and 1.25 GHz respectively. The setting frequency of pump pulse has been corrected (line 120).

3. The correlation denoising algorithm is based on the assumption that the spectrum is Lorentz shape added with uncorrelated noise. The correlation of signal and a standard Lorentz shape without noise can clear the uncorrelated noise. Since the correlation of two Lorentz shape is still a Lorentz shape, iterative correlations can further denoise the signal. While the correlation will broaden the width of Lorentz shape and increase the frequency uncertainty, we take the proper iterative time as two (line 238-240).

Reviewer 2 Report

Describe in much better detail the equipment and method used to build the experimental setup.

Also describe with better detail the references for the polarization management block.

Compare the advancements obtainable with the proposed methodology against the performance increase obtainable with other methods cited

Add pictures of the experimental setup.

Author Response

Thank you for your opinions for our work.

1.The type and model of equipment used to build the experimental setup (line 157, 160, 168, 198) and more detail of methods has been added.

2.The polarization dependency of simulated Brillouin scattering lead to intensity fluctuation of the Brillouin gain. To eliminate the polarization dependency, an equi-power pump array on both orthogonal directions of polarization is introduced. The intensity of Brillouin gain would be stable no matter the probe light is in which direction of polarization. To make the intensity of pump equal on both orthogonal directions of polarization, we duplicate the pump by a polarizing beam splitter (PBS), and combine them with a polarizing beam combiner (PBC) with delay on one direction of polarization as shown in Figure 4 to separate them in time domain. We detect the output of PBC by a photon detector and oscilloscope and tune the polarization controller to obtain the identical power.

3.There are several proposed BOTDA scheme that can detect about 10 kilometers fiber link within 1 ms as far as we know. 

The first one is digital optical frequency comb (DOFC) based BOTDA, whose spatial resolution reaches 12.5 m and frequency uncertainty is 2 MHz by utilizing multiple pump pulses. However, it is very difficult to obtain an identical power of several pump pulses because of the power saturation of EDFA, leading to failure of solving the correct Brillouin frequency shift. Our proposed scheme is also based on DOFC, whose spatial resolution reaches 10.24 m and frequency uncertainty remains in 2 MHz. Besides, since the pump is qusai-continuous, the pump array can get an identical power much more easily. 

The second one is optical chirp chain (OCC) based BOTDA, which can achieve spatial resolutions of 2 m with 6.25 MHz sampling rate but work on short fiber (50 m long). Also, it can be used on long range detection (150 km) by utilizing Raman amplifier, with sacrificing the sensing speed due to large amount of averaging. Besides, the frequency detecting dynamic range OCC-BOTDA is few hundreds of MHz, much smaller than our proposed scheme (over 2 GHz) because of the frequency span of probe light.

4.The complete experimental setup has been shown in Figure 4.

Reviewer 3 Report

In this paper, the authors present a resolution and SNR improvement scheme for DOFC based BOTDA ultrafast sensing system. With a new signal process model based on the convolution theorem and pulse coding method, experimental results indicate the spatial resolution reaches 10.24 m and the frequency below 2 MHz over 9.5 km long fiber. The method present here can find applications in FPGA-based long-time dynamic measurement. This article is clear, concise, and suitable for the scope of the journal. Only several small suggestions:

1.      the 3 port of one circulator of the experiment scheme in the Fig.4 is not connected, please check.  

2.      Suggest the authors improve the resolution of Fig.6.

3.      The overlapping issue of pulse array could be explained more clearly.

Author Response

Thanks for your opinions for our work.

1.The optical circulator after the optical isolator has the same effect of the isolator, which can stop the proceeding pump pulses. The 3 port of this optical circulator is blocked. The figure has been edited to avoid the confusion (line 188).

2.Figure 6 has been edited as request (line 229).

3.The orthogonal directions of polarization of optical pump pulse should be separated in time domain in order to obtain an identical power and avoid the power fluctuations. If a non-return-zero PRBS pump array passes the polarization management part in Figure 4, the duplicated pump of the former bit will overlap with the latter bit.

Reviewer 4 Report

The manuscript is well written. The authors can include the following:   1. details of digital pulse processing algorithm (which is used to solve the impulse response matrix) implementation and hardware details.   2. Also, authors can specify overall latency in the complete measurement, which will provide an estimation of timing requirements.  

Author Response

Thanks for your opinions for our work.

1.The equation (4) shows the relationship among the input, output and impulse response matrix after 2D-FFT (or we can say in frequency domain, although this "frequency" is generated by 2D-FFT instead of the real frequency). For a linear invariant system, the product of input and impulse response matrix is equal to output matrix in frequency domain. Distinguished from matrix multiplication, the product on the left side of equation is performed by multiplying element by element, called Hadamard product. So it is simple to perform a inverse operation, i.e. devide element by element, to obtain the impulse response matrix in frequency domain, which is noted as shown "A./B" in figure 2. More details have been added in part 2 (line 102-114). Besides, the model and type of experimental setup is added (line 157, 160, 168, 198).

2.The overall latency of one measurement is the period of pump array, which is 209.6 microseconds in this paper. With the detection of probe within the same time, the output matrix is completed and impulse response matrix can be solved. Since the pump array is qusai-continuous, there is no gap between the adjacent period.